# “I Solemnly Swear”: A Comparative Study of Codes of Professional Ethics amongst Pharmacists from Culturally Diverse European Countries

**DOI:** 10.3390/pharmacy12050143

**Published:** 2024-09-24

**Authors:** Raquel Raimundo, Afonso Cavaco

**Affiliations:** 1Faculdade de Farmácia, Universidade de Lisboa, Av. Prof. Gama Pinto, 1649-003 Lisboa, Portugal; 2Departamento de Farmácia, Farmacologia e Tecnologias em Saúde, Faculdade de Farmácia, Universidade de Lisboa, Av. Prof. Gama Pinto, 1649-003 Lisboa, Portugal

**Keywords:** ethics, professional codes, professional autonomy, pharmacy practice, interprofessional relations

## Abstract

Ethical practice is a universal concern for healthcare professionals, independent of their social, cultural, or religious background. This study aimed to assess and categorise statements published in codes of ethics for pharmacists from three diverse societies within the wider European area. The study followed a qualitative exploratory and triangular design, comparing the leading professional and ethical statements between three geographically apart countries (Portugal, Lithuania, and Turkey) and using the International Pharmaceutical Federation Code of Ethics proposal as a gold standard. Common core values such as honesty, integrity, and professional autonomy were identified across the countries’ codes, suggesting that shared recognised core values underpin pharmacists’ practice and policies in culturally diverse settings. None of the codes fully correspond to the framework the International Pharmaceutical Federation proposed. The analysis elicited significant inconsistencies between the codes for analogous practice models within the same continent. Further studies are needed to gain a more profound and comprehensive understanding of the underlying reasons for these discrepancies so that ethical weaknesses can be improved and harmonisation towards best-practice principles can benefit patients and healthcare systems.

## 1. Introduction

Over time, the pharmacy profession has undergone significant restructuring, creating new duties and competencies [1]. Today, pharmacy is still a profession in transition [2], characterised by a marked shift from a product-based model to more person-centred care [3]. For this shift to be successful, there must be a unanimous and continuous social and political drive on the role of pharmacy in society, where policymakers view pharmacies as an essential part of healthcare provision, bringing additional practice and ethical requirements to pharmacists [4].

Although pharmacy has traditionally been diverted from the bioethical debate, pharmacists face numerous ethical dilemmas daily [5]. This lack of discussion leads to a lack of recognition of new issues shaping pharmacy practice, such as the corporatisation of pharmacies and the increasing pressures professionals face [6,7]. The tension between strict professional conduct and business expectations creates an environment of divided loyalties, leading to moral issues for pharmacists in different latitudes [8]. Additionally, an increasing demoralisation and a lack of professional pride and identity are observed, especially amongst young pharmacists who are often disillusioned with the reality of pharmacy practice [9].

Codes of ethics play a critical role in this context. They aim to enshrine the profession’s fundamental values in a single document while promoting ethical behaviours and safeguarding professional status and the right to professional autonomy [10,11].

The International Pharmaceutical Federation, originally Fédération Internationale Pharmaceutique (FIP), has long highlighted the relevance of pharmacy ethics. The FIP has published a statement on professional standards, including recommendations that should be incorporated into existing ethical codes for pharmacists worldwide. The FIP has also reaffirmed and publicised the commitments that form the basis for the roles and responsibilities of pharmacists worldwide [12].

The regulations of pharmaceutical activity can vary significantly depending on the country. For instance, in Portugal, pharmacies are no longer exclusively owned by pharmacists, although pharmacists in service are mandatory [13]. Deontologically, the Portuguese Pharmaceutical Society (the registration body) has the right to take disciplinary action, meaning the published code of ethics applies to all working pharmacies [14]. In countries from other European geographies, such as Lithuania, pharmacy ownership is also not restricted to pharmacists, while in Turkey, only Turkish citizens with a diploma from a pharmaceutical faculty can open a pharmacy [15].

In Lithuania, several pharmaceutical professional associations are active [16], there is a fragmented pharmacists’ representation, and there was no unified code of ethics until recently (changes operated in 2023). The new code, which is not legally binding, summarises the ethical guidelines considered necessary by Lithuanian pharmacists and aims to follow the recommendations made by the FIP.

In Turkey, the Turkish Pharmacists Association (TPA) is divided into 54 regional chambers [16] and undertakes tasks related to establishing and implementing professional and ethical rules for pharmacy. The Turkish Code of Professional Ethics for Pharmacists was published in 1968 (presently under revision), and to ensure compliance, chambers of pharmacists have disciplinary committees, while the TPA has a High Disciplinary Committee [17].

Considering the constant need to advance pharmacists’ ethical and moral standpoints in a rapidly evolving and demanding practice environment, a thorough comparative study of the codes of ethics in different countries opens the door to confirming core and shared ethical values defining the profession’s scope and societal objectives. The degree of alignment between the several national professional codes belonging to common geography, including their similarities and differences, will provide a clearer picture of an overall profession’s sense of moral consciousness and degree of professionalism.

## 2. Materials and Methods

This study used a documentary qualitative thematic analysis. Analytical support was obtained through free qualitative data analysis (QDA) software (the QualCoder application, version 3.2) [18] to examine the documents containing different codes of ethics.

Three countries from the European region were purposefully selected to compare different pharmaceutical codes of ethics. The criteria used for the selection were geographical and sociocultural disparities, while similarities comprised healthcare provision, especially concerning the functioning of pharmacies. Access to legal information, deontological norms prevailing in the pharmaceutical practice of selected countries, and inclusion or exclusion criteria were also considered.

The three countries from the European region purposefully selected to compare different codes were Portugal, Lithuania, and Turkey. The criteria used for the selection were geographical separation and sociocultural differences, while similarities comprised healthcare provision, especially concerning the functioning of pharmacies. Access to legal information and deontological norms prevailing in the pharmaceutical practice of selected countries also determined the selection. A variation in the binding status from the European Union (EU) regulations for Portugal and Lithuania, promoting normative proximity between these countries, compared to Turkey as an EU external country, was considered an advantage.

Differences in social background were expected to provide specificities from south Mediterranean and Nordic–Baltic practice contexts, particularly for Turkey, knowing the European and Asian sociocultural influences and the dominance of Islam, countering the Christian moral background in Portugal and Lithuania.

The corresponding pharmacists’ codes of ethics were obtained from the official organisations and through the collaboration of academics and/or experts from each country.

All documents were submitted by these accredited contact persons, who are familiar with the regulations of the pharmacy profession in their country and the existing codes of ethics.

Regarding data extraction, the FIP statement [12] was first uploaded to the QDA software, and individual statements were exhaustively and thematically coded until saturation.

The resulting framework was later used to analyse each country’s code of ethics sections, consistent in content with the coding frame generated from the FIP.

The remaining text segments that could not be classified using the FIP coding frame were subsequently analysed through open coding.

The Turkish and Lithuanian codes of ethics were translated into English using free online translation services. To address any issues with translation accuracy and interpretation, the research team reviewed sentences that raised concerns about readability or meaning and subsequently discussed these with academics and experts from Turkey and Lithuania. Being native speakers, these scholars translated the passages directly into English and solved any pending interpretation doubts. The research team also translated the Portuguese code of ethics into English to fit the coding frame.

The resulting codes were sorted into hierarchical groups based on the proximity of keywords and concepts with the FIP coding frame. Upon iterative coding and tree saturation, it was possible to review internal relationships within minor themes of codes and derive an overall coding map. This code categorisation and linkage contributed to an overarching theme that enriched the graphical representation of the coding.

## 3. Results

After completion of coding the documents from the FIP, Portugal, Lithuania, and Turkey, six general coding themes were created, namely “A—Economic matters”, “B—Genetic ethics”, “C—Pharmacist-patient relationship”, “D—Relationship with the Pharmaceutical Society”, “E—Professional positioning”, and “F—Interprofessional cooperation”. Several subthemes and the corresponding coding characterise each of these main themes.

Figure 1 presents the conceptual map with all the codes and themes developed, with those from the FIP marked accordingly.

From the qualitative analysis of the FIP document, 24 different codes emerged. Each code was formulated by analysing the individual premises. Table 1 shows the codes mentioned above and the corresponding FIP text segment.

The open coding of the ethical documents from the three selected countries resulted in 41 new codes (beyond the FIP), which were used to code 108 statements from these documents. Since some codes could be considered allusive in their meaning or allow for variable interpretation, a description is presented to facilitate their understanding. Others were deemed self-explanatory or were too narrowly defined to lead to second interpretations. Table 2 shows the code detailing that could generate doubt; all coding details can be found in the Appendix A.

### 3.1. Portugal

In total, 53 codes were identified, summing 193 code counts, including repetitions. This code of ethics was the most extensive, with 57 articles and over 5000 words. Figure 2 illustrates the frequency of the most applied codes in the Portuguese code of ethics.

All FIP codes were found, except for the codes “C.3.1—Spirit of collaboration, with patients and carers”, “C.4.1.2—Respecting divergence, caregivers”, E.4.3.2—Managing loyalties”, and “F.2.2.1—Respecting divergence, healthcare professionals”, meaning that 20 were present.

Regarding the frequency of codes by theme, “E—Professional positioning” has the highest relative frequency (26%), and all the themes were used in the codification. 

The Portuguese code of ethics was the only one to address pharmacogenetics, hypothetical war situations, and prisoner health services issues. It also emphasises the need to comply with other healthcare professionals, pharmacists, and healthcare industries. Regarding the need for interprofessional collaboration, it is essential to note that while the emphasis is on collaboration, there is no mention of the possibility of divergence between professionals, whether moral or clinical. Although the possible conflict of loyalties (to the patient and the employer) is not recognised, recognition of and emphasis on professional autonomy are clear. Workers’ rights also have their say in the Portuguese document, where the right to a fair wage, non-discrimination policies, strikes, and the defence and preservation of professional dignity are defended. The pharmacists’ moral and clinical objections are recognised.

The text also addresses patient registers, emphasising the importance of data protection, the need for patients’ informed consent, and the freedom of the information collected. Regarding the need for patient respect, the Portuguese code of ethics stresses no discrimination by race, sex, gender, genetic constitution, race, ethnicity, language, region of origin, religion, political or ideological beliefs, economic situation, social situation, state of health, disability, age or sexual orientation, and type of illness. An illustrative map that lists all the codes used in the Portuguese code of ethics can be found in the Appendix A.

### 3.2. Lithuania

This document used 23 codes for a total of 35 codes. It consists of five articles, for a total of about 600 words. Figure 3 illustrates the frequency of the most applied codes in the Lithuanian code of ethics.

Analysing the code frequency for FIP codes, the following codes were not present: “C.1.1.3—Confidentiality, compliance with the law”, “C.1.1.4—Confidentiality, informed consent”, “E.2.3—Equity”, “E.2.4—Justice”, “E.2.5—Resource managing”, “E.4.3.2—Managing loyalties”, and “F.1.1—Spirit of collaboration, with healthcare industries”. This means that from the 24 FIP codes, 17 were identified.

As for the relative frequency of the coded themes, “F—Interprofessional cooperation” has the highest relative frequency (48%). The themes “A—Economic matters”, “B—Genetic Ethics”, and “D—Relationship with the Pharmaceutical Society” were not present. 

The Lithuanian code greatly emphasises professional cooperation, especially with other pharmacists, regarding the need for patient respect. It stresses reasons not to discriminate, as follows: age, nationality, religion, gender, education, wealth, and minority status. Regarding non-discrimination policies, it is noticeable that the Lithuanian code, unlike the others analysed, does not contain provisions on sexual orientation. An illustrative map that lists all the codes used in the Lithuanian code of ethics can be found in the Appendix A.

### 3.3. Turkey

The Turkish document presented 24 codes, summing up to 33 total codes count. This document has 20 articles and over 900 words. Figure 4 illustrates the frequency of the most applied codes in the Turkish code of ethics.

Analysing the code frequency from the FIP, the following were not identified: “C.1.2.4—Confidentiality, informed consent, “C.4.1.2—Respecting difference, caregiver”, “C.4.1.3—Continuity of care”, E.2.3—Equity”, “E.2.4—Justice”, and “F.2.2.1—Respecting divergence, healthcare professionals”. This means that of the 24 FIP codes, 18 were present.

The coding frequency by theme was reasonably balanced, although “B—Genetic ethics” was absent. The Turkish code of ethics was the only one that mentioned animal health and veterinary medical products as a task and responsibility of pharmacists. Although the need for collaboration between pharmacists and other health professionals is mentioned, there is no reference to expected attitudes if divergence emerges from interprofessional collaboration. However, the role of the pharmaceutical society in delegating conflicts between pharmacists is mentioned when such intervention is required.

The Turkish pharmaceutical code of ethics states that pharmacists must not show patient discrimination, regardless of gender, language, race, nationality, philosophical belief, religion, moral opinion, character and personality, social class, or political beliefs. Similar to the Lithuanian code, there is no mention of the need for non-discrimination based on sexual orientation. An illustrative map that lists all the codes used in the Turkish pharmaceutical code of ethics can be found in the Appendix A.

Figure 5 illustrates the frequency of each theme regarding the three national codes of ethics.

## 4. Discussion

This study aimed to use an internationally accepted ethical framework as a guide to compare the professional ethical codes of three geographically distant and culturally diverse European countries. Although it would be challenging to produce a single transnational code, even within the EU area, given the different legal frameworks in different countries, this study has shown that improvements can bring codes of ethics closer to the FIP standard [12].

Essential ethical values run through all the ethical documents analysed, regardless of cultural background and specific deontological practices, such as honesty, integrity, professional autonomy, and patient safety and welfare, aligning with high-hierarchical ethical principles. Unique ethical issues and priorities were also found, such as the importance of patient confidentiality and the protection of the pharmacist’s professional role in Portugal. In Turkey, on the other hand, the dependence on physicians and the protection of animals is a significant theme. Even with the main themes, the countries show different priorities. While “E—Professional positioning” was frequently used in the Portuguese and Turkish codes, the Lithuanian code underlined the theme “F—Interprofessional collaboration”. This demonstrates the natural need to adjust each code to the main concerns and needs of the situated professions it serves.

The codes were created and approved in different periods. This could explain, for example, the lack of information on pharmacogenetics and non-discrimination based on sexual orientation. However, this applies well to the Turkish code since Lithuania has a recent code of ethics. The difference in oldness between codes cannot also explain why the Portuguese and Lithuanian documents do not include topics on animal welfare, while the Turkish code, the oldest of the three, clearly mentions this topic.

The Turkish code also reveals a vision of a more restrictive professional activity and a greater emphasis on complying with recognised rules. The limits of professional practice are clearly emphasised, namely the rejection of diagnoses and the dependence on doctors since it is not allowed to change or administer prescribed medicines. For instance, vaccine administration by pharmacists is a new emerging trend, with particular interest from the FIP, and is widely implemented in Portugal [26], although it is prohibited in Turkey [27]. The prohibition of advertising medicines is also mentioned, with very strict exceptions for what is allowed. In summary, the abovementioned restrictions and the constant use of the “need to comply” illustrate a punishment-centred code of ethics compared to the other two codes analysed. This fact could be justified by the deontological context of the TPA and its actual legal authority and enforcement capacities. Although it was impossible to confirm the extent of enforcement from the national organisations, the lighter emphasis on compliance with mandatory laws and norms suggests a more voluntary adoption by professionals in Portugal and Lithuania than in Turkey.

The codes of ethics analysed often fall into a broader spectrum of moral reasoning and may fail to address objective dilemmas and moral predicaments that usually plague the pharmacists’ professional lives, such as market and sales indicators impregnated with unreasonable consumerism. This lack of directness takes on even more significance when pharmacists often adopt a posture of ethical passivity and fail to reconstruct the moments of necessary ethical decision-making [28]. This weakness is also objectively reflected in the fact that none of the ethical documents studied mention the possible existence of moral distress due to a sense of dual loyalty, considering that pharmacists often play the role of serving society and, at the same time, their employers and the profit ideals of the company for which they work. This issue is critical given the current trend towards corporatisation of community pharmacies. While many European countries had laws that tied ownership of pharmacies exclusively to pharmacists, some countries have chosen to move to a system that allows corporative and unspecified ownership. This phenomenon exacerbates a long-standing dilemma for pharmacists, i.e., how to cope with economic and professional pressures simultaneously. This setting creates an environment of divided loyalties, leading to tensions and moral problems for the practitioner [8]. However, it is essential to note that the lack of professional pride and identity amongst younger pharmacists is addressed in the ethical codes analysed, as all point to the importance of mentoring and camaraderie amongst co-workers and peers [29].

From the several ethical principles, the need for an improvement in recognition of professional autonomy is another theme that stands out: it is remarkably defended in all the ethical codes studied. This is particularly important as the pharmacist profession’s rights are considered a current and relevant matter that should be remembered and taken seriously. For a profession to defend its rights, it is necessary to know what these rights consist of. The Portuguese, Lithuanian, and Turkish codes of ethics ensure these principles remain present and affirmed for the pharmaceutical profession.

In addition to similarities and differences between countries and their pharmacists’ codes of ethics, the FIP’s ethical framework needs to be fully integrated into the national codes. Although the FIP framework cannot be assumed to be a fully operational code, it addresses the universal themes that shape what is presently understood as a humanistic and moral-bound practice, regardless of social or cultural differences. The full endorsement of the FIP means one step closer to unifying the ethical discussion on pharmaceutical practice and achieving a more comprehensive vision of pharmacists’ professional roles and responsibilities towards patients, communities, and societies.

This study presents limitations. The coding map was based on the FIP framework and three national codes of ethics, which prevents the results from being extrapolated to other European countries or worldwide. Other limitations comprise language and translation inaccuracies, particularly for the Lithuanian and Turkish codes; specific details could have been lost, even when native speakers checked ambiguous translated quotations. Although an official translator is always preferable, there was no budget for translation services available. Some original sentences and their translation were introduced in the Appendix A as examples of the translation followed.

Another possible limitation is that one coder conducted the documentary analysis, with interpretation issues discussed with a second researcher. This process may have led to missing a broader set of alternative interpretations and perspectives. Finally, it can be argued that this analysis was more focused on the work of community pharmacists, knowing that codes of ethics apply to all pharmacists, regardless of their professional role; however, this study was primarily aimed at pharmacists whose work involves direct contact with patients.

## 5. Conclusions

This study contributed to gaining new comparative information and reflections on the structure and content of three different codes of ethics in greater Europe. It also allowed an active comparison between actual codes and a recognised framework for pharmacy ethics and conduct, showing there are still missing parts in implementing and adapting to the deontological practice context in different countries.

The FIP framework is a clear starting point that emphasises the fundamentals of ethics in pharmacy and serves as a good starting model for countries developing a code of conduct for the pharmaceutical sector. However, an analysis of three national codes shows that the FIP still needs topics that ensure a richer and expanded world of ethics in pharmacy practice.

Since professional organisations are moral actors in a self-organised and autonomous profession, their influence on the professionals’ moral judgments and ethical trends should not be neglected. This perspective is crucial to provide the proper ethical support to the professionals they support, thus indirectly ensuring more comprehensive and higher-quality patient care. In this sense, the findings of this study could serve as a guide for professional organisations to revise and update their codes, in particular, to include a treatise on general ethical concerns related to pharmacy practice, such as the moral burden of dual loyalty and the need to strive for equity and justice in the distribution of medicines and therapies, amongst other pressuring issues.

In the future, with a more thorough validation of the code tree obtained, this study could also serve as a basis for analysing other European pharmaceutical codes of ethics to develop the proposed mapping and formulate new links between the data collected.

## Figures and Tables

**Figure 1 pharmacy-12-00143-f001:**
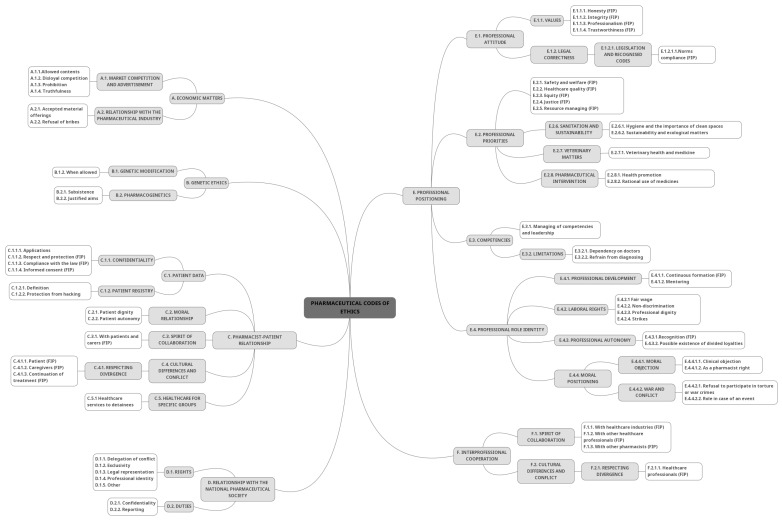
Conceptual map with all codes.

**Figure 2 pharmacy-12-00143-f002:**
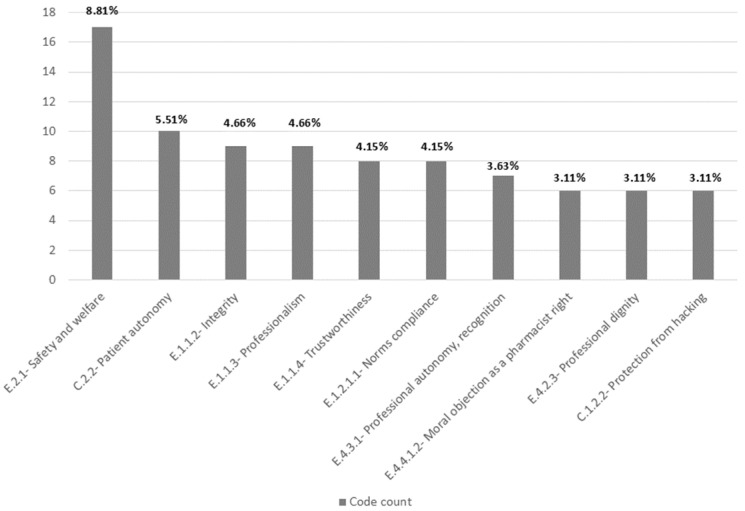
Bar chart of code count and frequency—Portugal.

**Figure 3 pharmacy-12-00143-f003:**
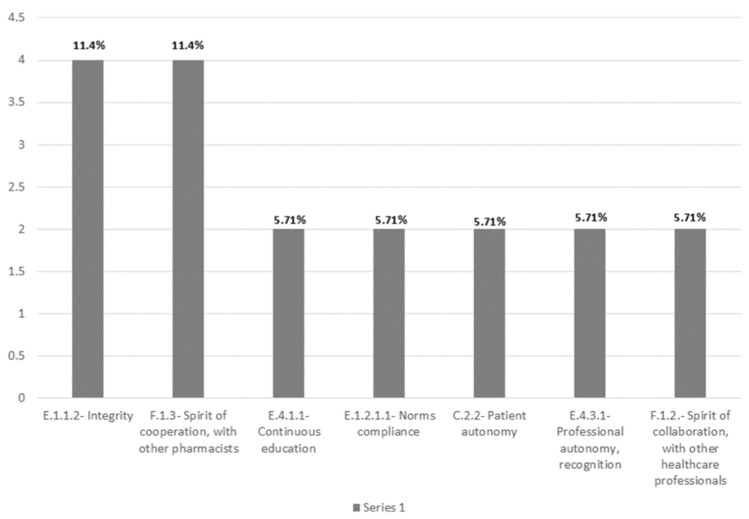
Bar chart of code count and frequency—Lithuania.

**Figure 4 pharmacy-12-00143-f004:**
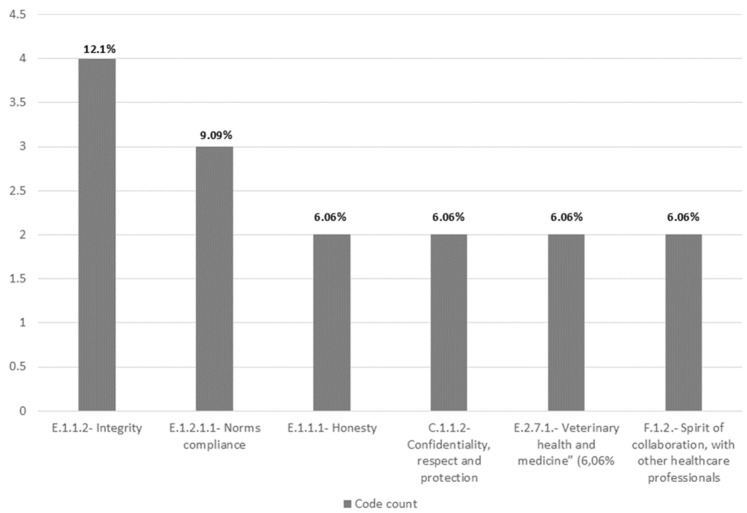
Bar chart of code count and frequency—Turkey.

**Figure 5 pharmacy-12-00143-f005:**
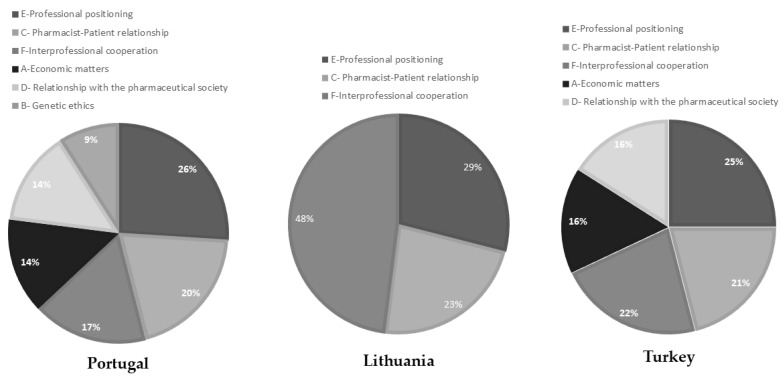
Pie chart of counts of codes for each country.

**Table 1 pharmacy-12-00143-t001:** Resulting codes of the FIP code of ethics.

Codes	Sub-Codes	Statements
C.1. Patient data	C.1.1.2. Respect and protection	“To respect and protect the confidentiality of patient information acquired or accessed while providing professional services (…)”.
	C.1.1.3. Compliance with the law	“(…) as allowed by applicable legislation and regulation”.
	C.1.1.4. Informed consent	“(…) to ensure that such information is only disclosed with the informed consent of that individual (…)”.
C.3./F.1. Spirit of cooperation	C.3.1. With patients and carers	“To cooperate and collaborate with colleagues, other health professionals, consumers, patients, carers, and other actors in the healthcare delivery system”.
	F.1.1. With healthcare industries	
	F.1.2. With other healthcare professionals	
	F.1.3. With other pharmacists	
C.4.1./F.2. Respecting divergence	C.4.1.1. Patient	“To respect patients’ rights and recognise and respect the cultural differences, beliefs and values of patients, carers and other healthcare professionals (…)”.
	C.4.1.2. Caregivers	
	F.2.1. Healthcare professionals	
	C.4.1.3. Continuation of treatment	“To ensure continuity of care for the patient in the event of a conflict with their own moral or religious beliefs based on respect for patient autonomy”.
E.1.1.1. Honesty/E.1.1.2. Integrity		“To act with honesty and integrity”.
E.1.1.3. Professionalism		“Always act professionally, by scientific principles and professional standards, including those developed by the International Pharmaceutical Federation”.
E.1.1.4. Trustworthiness		“(…) not engage in any behaviour or activity likely to bring the profession into disrepute or to undermine public confidence in the profession”.
E.1.2. Legal correctness	E.1.2.1.1. Norms compliance	“To comply with legislation and accepted codes and standards of practice in the provision of all professional services and pharmaceutical products and to ensure the integrity of the supply chain for medicines”.
E.2.1. Safety and welfare		“To ensure that their priorities are the safety, well-being and the best interests of those to whom they provide professional services”.
E.2.2. Healthcare quality		“(…) to ensure that the best possible quality of healthcare is provided both to individuals and the community at large (…)”.
E.2.3. Equity/E.2.4. Justice		“(…) and the principles of equity and justice”.
E.2.5. Resource managing		“(…) while always considering the limitations of available resources (…)”.
E.4.3. Professional autonomy	E.4.3.1. Recognition	“(…) act at all times as autonomous health professionals (…)”.
	E.4.3.2. Managing loyalties	“(…) recognizing the challenges posed by divided loyalties and the potential in many settings for conflicts of interest that need careful management”.
E.4.4.1. Continuous education		“To ensure that they maintain competence through continuing professional development”.

**Table 2 pharmacy-12-00143-t002:** Codes detailing.

Codes		Subcodes	Description
A.2. Relationship with the pharmaceutical industry		A.2.1. Accepted material offerings	Possible exceptions to the definition of bribery include what can be classified as non-bribery.
		A.2.2. Refusal of bribes	Pharmacists must not accept payments or incentives from the pharmaceutical industry that could mislead them and unethically alter the health services they provide.
B.2. Pharmacogenetics		B.2.1. Subsistence	Mention the existence of pharmacogenetics and the reality of its application in the current era of pharmacological development.
C.1.2. Patient registry		C.1.2.1. Definition	What do patient registers consist of, and what is their purpose.
		C.1.2.2. Protection from hacking	Obligation to take measures to prevent third parties from accessing patient data.
D. Relationship with the pharmaceutical society	D.2. Duties	D.2.1. Confidentiality	Professional regulation may classify some information as confidential; pharmacists must respect this confidentiality and not disclose the information to the public.
		D.2.2. Reporting	The reporting of violations of the law or other acts affecting the ethical and deontological aspects of the pharmaceutical profession.
	D.1. Rights	D.1.1. Delegation of conflict	In case of conflicts between pharmacists, the pharmaceutical society can act as a moderator and help to settle disputes.
		D.1.2. Exclusivity	Only pharmacists (who are registered in the pharmaceutical society) can open pharmacies.
		D.1.4. Professional identity	Members have the right to call themselves pharmacists because they have a university education in pharmaceutical sciences and are members of the pharmaceutical society of their country.
E.3. Competencies		E.3.1. Managing competencies and leadership	Pharmacists need to be aware of their skills and limitations in their work and, simultaneously, recognise the skills and limitations of other pharmacists and health professionals to lead and work together.
		E.3.2.1. Dependency on doctors	Pharmacists cannot change doctors’ prescriptions or administer medicines to patients.
E.4.1. Professional development		E.4.1.2. Mentoring	Pharmacists should take an active role in the formation of future professionals.
E.4.2. Laboral rights		E.4.2.2. Non-discrimination	The pharmacist shall not be discriminated against based on a belief or personal practice.
E.4.3. Professional autonomy		E.4.3.1. Professional autonomy	Professions can be distinguished by several unique characteristics that assign them specific functions, social structures, representations, and forms of knowledge. This gives professionals exclusive and in-depth skills and abilities usually acquired in specialised institutions [19]. By this demarcation by specific and certified expertise, the profession may have the prerogative of professional autonomy, reflected in the privilege of self-regulation and materialised in a particular code of ethics [20].
		E.4.3.2. Managing loyalties	While pharmacists have a covenant with patients and a moral and professional obligation to serve their best interests, pharmacy is also a business that must be profitable [21]. This creates an environment of divided loyalties, leading to tensions and moral distress for the pharmacist. The term is defined as a mental and physical state experienced due to limitations in professional practice, resulting in having to perform an act perceived as ethically wrong [22,23].
E.4.4. Moral positioning		E.4.4.1. Moral objection	Conscientiousness is a form of self-reflection that leads to a particular judgement that guides a person’s actions [24]. Conscientious objection is a refusal to perform a particular act predicted by law based on the person’s moral beliefs [25]. In healthcare, the conscious objection is refusing certain services and treatments [24].

## Data Availability

Primary data is available from the Portuguese Pharmaceutical Society, the Lithuanian Pharmaceutical Association and the Ankara Chamber of Pharmacists.

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
