# Peer review of "“I Solemnly Swear”: A Comparative Study of Codes of Professional Ethics amongst Pharmacists from Culturally Diverse European Countries"

_pharmacy, 2024, doi:10.3390/pharmacy12050143_

Round 1
Reviewer 1 Report
Comments and Suggestions for Authors
The considerations are presented in the file attached to the review.

Author Response
Dear Reviewer 1.
The authors would like to thank you for all the valuable suggestions. These were accordingly placed in the revised manuscript, and changes were highlighted.
Thank you very much.
Reviewer 2 Report
Comments and Suggestions for Authors
The paper compares the Codes of Ethics for pharmacists from Portugal, Lithuania, and Turkey, using the International Pharmaceutical Federation (FIP) Code of Ethics as a benchmark. The study's qualitative exploratory design highlights the core values shared across these diverse cultures, such as honesty, integrity, and professional autonomy, while also identifying significant inconsistencies between the national codes and the FIP framework.
Strengths
Comparative Approach: The study's comparative methodology provides valuable insights into how different cultural contexts influence ethical standards in pharmacy practice.
Use of FIP Framework: Employing the FIP Code of Ethics as a benchmark allows for a standardized comparison and highlights areas where national codes can improve.
Detailed Analysis: The paper offers a thorough analysis of the codes, including the frequency and context of various ethical themes.
Weaknesses
Limited Scope: The study focuses only on three countries, which may limit the generalizability of the findings to other European nations or globally.
Translation Issues: The reliance on translation services for the Lithuanian and Turkish codes may have introduced inaccuracies despite efforts to mitigate this with native speakers' input.
Questions for the Authors
Why were Portugal, Lithuania, and Turkey chosen explicitly for this study, and how might adding additional countries have impacted the findings?
The paper mentions ethical passivity among pharmacists. Could you provide more context or examples of how this manifests in practice and its implications?
While professional autonomy is emphasized, how do the codes address potential conflicts between professional independence and business pressures, especially in corporatized pharmacy settings?
What are the next steps for research in this area, and how can future studies build on your findings to further harmonize ethical standards in pharmacy practice?
By addressing these questions, the authors can provide further clarity and depth to their study, enhancing its contribution to the field of pharmacy ethics.
Comments on the Quality of English LanguageEnglish is fine; only minor editing is required.
Author Response
Dear Reviewer 2.
The authors would like to thank you for the valuable comments. Please find our replies in the attached file.
Thank you very much.

Round 2
Reviewer 1 Report
Comments and Suggestions for Authors
I have reviewed the revisions made by the authors and find that all necessary changes have been addressed. The article now meets the required standards and is ready for publication.